# A Novel Localization Technique Using Luminous Flux

**Muhammad Irshad [1], Wenyuan Liu [1],\*, Jehangir Arshad [3], M. Noman Sohail [1], Aparna Murthy [2], Maryam Khokhar [4] and M Musa Uba [1]**

[1] School of Information Science and Engineering, Yanshan University, Qinhuangdao 066004, China; ibrahim@stumail.ysu.edu.cn (M.I.); mn.sohail@stumail.ysu.edu.cn (M.N.S.); musaubamuhammad@gmail.com (M.M.U.)

[2] EIT, PEO, Toronto, ON M3C 1X5, Canada; aparnasm11@gmail.com

[3] Electrical and Computer Engineering Department, COMSATS University Islamabad, Lahore Campus, Lahore 54810, Pakistan; jehangir@cuisahiwal.edu.pk

[4] School of Economics and Management, Yanshan University, Qinhuangdao 066004, China; maryamkhokhar@stumail.ysu.edu.cn

**\*** Correspondence: wyliu@ysu.edu.cn; Tel.: +86-180-3166-1988

**Abstract:** As global navigation satellite system (GNNS) signals are unable to enter indoor spaces, substitute methods such as indoor localization-based visible light communication (VLC) are gaining the attention of researchers. In this paper, the systematic investigation of a VLC channel is performed for both direct and indirect line of sight (LoS) by utilizing the impulse response of indoor optical wireless channels. In order to examine the localization scenario, two light-emitting diode (LED) grid patterns are used. The received signal strength (RSS) is observed based on the positional dilution of precision (PDoP), a subset of the dilution of precision (DoP) used in global navigation satellite system (GNSS) positioning. In total, $31 \times 31$ possible positional tags are set for a given PDoP configuration. The values for positional error in terms of root mean square error (RMSE) and the sum of squared errors (SSE) are taken into consideration. The performance of the proposed approach is validated by simulation results according to the selected indoor space. The results show that the position accuracy enhanced is at short range by 24% by utilizing the PDoP metric. As confirmation, the modeled accuracy is compared with perceived accuracy results. This study determines the application and design of future optical wireless systems specifically for indoor localization.

**Keywords:** positional dilution of precision (PDoP), localization; indoor positioning; received signal strength (RSS), visible light communication

## 1. Introduction

Indoor positioning systems are of great interest to researchers due to the need to deliver services (i.e., store mapping, routes, floor-associated pinning) to users according to their locations. Global navigation satellite systems (GNNS) are widely implemented around the globe. However, GNNS signals are unable to penetrate through walls, specifically in indoor scenarios and urban canyons [1,2].

These indoor spaces and many urban canyons remain uncovered in existing location-based systems. Hence, numerous technologies have been proposed to address this issue in existing literature, including wireless fidelity (WiFi), ZigBee, Bluetooth, ultra-wide band (UWB), and radio frequency identification (RFID).

The problem of localization in indoor scenarios can also be solved by using visible light communication (VLC), which shows much better positioning accuracy. Generally, VLC-based indoor

localization is measured by using a visible light signal [3] that includes factors such as light-emitting diode **(**LED) positions, receiver orientation, light incident angles, and light arrival angles. All of the abovementioned factors are extracted from the light signal. Moreover, the VLC-based localization technique has two exclusive advantages.

(i) Optical frequencies in the visible light range from 430 THz to 770 THz can be easily regulated to avoid congestion, and this is a license-free range. Furthermore, the available bandwidth for communication in this range (i.e., 340 THz) is a thousand times larger than the combined 300 GHz bandwidth of radio waves and microwaves (i.e.**,** Ultra High Frequency UHF, Extremely High Frequency EHF, and Super High Frequency SHF) [4,5].

(ii) The second advantage is in terms of the multipath effect of optical signals that consume much lower power, as compared to the line of sight (LoS) signal. Both of the above advantages achieve the goal of this work by providing reasonably higher localization accuracy.

Currently, most of indoor applications are Wi-Fi-enabled. In the literature, it is the most studied technology used for indoor localization [6-9]. Horus is a WiFi-based localization software system [10] that relies on the fingerprinting technique. Moreover, authors in [10] have proposed a deep learning technique for training the data. In [11-21], authors have applied different localization techniques, including channel state information-based localization (CSI) [12,13], fingerprint-based localization [14–18], and additionally localization based upon angle of arrival (AoA), time of flight (ToF), and time difference of arrival (TDoA) [19–22]. All of these techniques are prone to multipath fading, clock synchronization, and environmental noise. However, existing Wi-Fi networks are efficient in getting the maximum data rate and network coverage rather than for localization purposes; for example, special hardware deployment is needed to read the signals containing localization information. In [22], authors used the received signal strength indicator (RSSI) in the online phase with a Kalman filter to observe the median accuracy.

Moreover, the room-level implication was performed in [23] by combining WiFi fingerprinting with a magnetic field, resulting 1.44 m localization error. Other technologies, such as UWB, acoustics, RFID, Bluetooth, and ultrasound, are also implemented in indoor tracking, positioning, and localization. Recently, authors applied UWB with moderate power consumption in [24]; however, it required extra hardware on different user devices. In [25], the authors presented the acoustics-based SITE, combining acoustic **S**ignals and **I**mages to achieve accurate and robust indoor loca**T**ion s**E**rvice, a technique that measures the difference in arrival angles corresponding to the different sources. Radio frequency identification (RFID) combined with pedestrian dead reckoning (PDR) was deployed in [26] to precisely estimate the motion of a user, however this depends on a common unit for reading RF and PDR. Another novel algorithm was proposed for Bluetooth technology in [27] that has a range of errors of roughly 2.4 m. Although the existing techniques used in this literature are novel, the abovementioned technologies are not widely deployable and are not suitable for extensive indoor positioning. Most of the prescribed techniques in the literature are useful for selective object identification, which may require special hardware infrastructure and additional costs.

Besides the aforementioned technologies, VLC-based localization techniques were also used by different researchers in [28,29]. The authors proposed an improved algorithm by applying machine learning techniques to retrieve localization information. More specifically, authors in [30] proposed a scheme for localization estimation by using the Lambertian emission pattern. In this technique, the user measurements from RSS are fitted into an interpreted propagation model. Thus, analytical results are limited to specific settings. A Recent study was published [31] on a technology that comprises infrared beacons, which measure positioning error for both LoS and Non-LoS in an indoor environment. However, this system is not widely deployed because it incurs an extra cost.

After a comprehensive study of the existing literature, we have proposed a novel approach to combine PDoP and RSS observations, and provide an explicit relationship between PDoP and the observable degree of root mean square (RMS) error by using the VLC communication channel. Secondly, to fully understand the higher-order terms of impulse response, single and multiple reflection measurements are performed. Additionally, in this study, the precedent estimation of

operation, particularly related to the required range, and consequent positional errors result in a propagation environment.

VLC channel models for various indoor environments, especially NLoS, are still in the exploratory phase. Authors in [32] stated that out of the total light, around 45% of light is received from NLoS interaction. Moreover, authors in [33] claimed that NLoS occurs in a dense cluttered room or space, resulting in multiple reflections from such objects or surfaces. Various NLoS scheme results significantly differ because of the lack of appropriate and general criteria to evaluate localization error.

The DoP has been used by different researchers for indoor positioning in estimating the positions of objects. In practice, it is divided into various elements, such as vertical DoP (VDoP), geometric DoP (GDoP), and positional DoP (PDoP) [34]. All of the abovementioned components are useful for evaluating the accuracy of a GNNS system. The details of DoP- and AoA-based positioning are comprehensively discussed in [35]. The authors evaluated radio signals to ensure the quality of an AoA position and determined those most affected by multipath. Further, the geometric dilution of precision (GDoP) with a machine learning algorithm was used to calculate the estimated value of GDoP in [36], in which the lower approximation error with fewer epochs was observed to estimate the position used by the mobile station. TDoA-based source location in three dimensions (3D) was analyzed with GDoP metrics in [37]. However, it was based solely on the deterministic location of nodes.

Here, we will apply VLC as a communication channel in an indoor localization method by using GNNS featuring PDoP, which is a subset of DoP. DoP relates to the measurement error in an indoor environment. This metric can be used to increase the accuracy [38]. The main goal is to investigate and analyze an indoor VLC channel model (for LoS and NLoS). Moreover, we will calculate and explore the impulse response (IR), signal-to-noise ratio (SNR), and the PDoP for RSS-based analysis in an indoor environment.

The influence of PDoP in a positioning system under the visible light spectrum is very limited. By using this metric, we can estimate the lower approximation error with fewer epochs in NLoS. This approach enables us to decouple the position propagation aspects for error estimation. Hence, we will investigate the PDoP attributes with our positioning model. Thus, this paper advocates that the PDoP approach be used to modify light sources and their positions randomly.

The potential contributions of this work can be summarized as follows.

- The configurations of the LED lattice are devised by using the Lambertian model for individual LEDs.
- According to our knowledge, no specific research has been performed using PDoP attributes prior to visible light-based indoor localization.
- We analyze PDoP with respect to channel impulse response time, considering both direct and multipath reflections, which has not been done before.
- Novel positioning algorithms are used to compute PDoP on each LED, along with a comparison of root mean square error (RMS) based on the simulated RSS values, the RMS errors, and the SSE errors are computed with two different LED lattices to estimate the positioning error.

The structure of this paper is as follows. Section 2 describes the system model (including the VLC principle) and the indoor channel models. Section 3 outlines the power measurement steps performed within the system. The adaptive collaboration positioning algorithm is discussed in Section 4. Experiments, results, and discussion are presented in Section 5. Finally, the concluding interpretations are given in Section 6.

## 2. System Configuration Method

The aim of this study is to investigate the impulse response of the VLC channel to exploit localization, which is done using confined luminaries. The luminaries, commonly called light-emitting diodes (LEDs), act as a transmission source for the light signal and the photodiode (PD), which collects information and acts as a receiver.

## 2.1. Working Principle of VLC Channel

The working principle of a VLC channel system is relatively simple—the data to be transmitted is mapped onto an electrical signal. The transmitters (LEDs) are driven by pulse width modulation orthogonal frequency multiplexing (PWM-OFDM). PWM-OFDM is a suitable candidate for achieving better frame synchronization in leading and trailing edges. On the receiver side, the detector uses a down-converter, through which the photocurrent is converted to measurable energy.

The channel model for indoor VLC systems is based on

$$y(t) = Rx(t) \otimes h(t) + \eta(t) \tag{1}$$

Here, R denotes the responsivity of the photodetector and $\eta(t)$ represents the Gaussian modeled noise; $\otimes$ expresses the convolution operation and $x(t)$ denotes the instantaneous input power, which cannot be negative ($x(t) > 0$). The impulse response acquired, $h(t)$, is convolved with $x(t)$.

The average power received at the photodetector as the photons impinge is given as

$$yP_\tau = H(0) * P_t \tag{2}$$

where $P_\tau$ denotes the average transmitted power that is an integral of $x(t)$ within the range $[-T,T]$ given as Equation (3), and $P_t$ is the total source power:

$$\lim_{T \to \infty} \left( \frac{1}{2T} \int_{-T}^{T} x(t)dt \right)^n \tag{3}$$

Moreover, the direct current (DC) gain of the channel is defined as:

$$H(0) = \int_{-T}^{T} h(t)dt \tag{4}$$

The channel modeling for indoor VLC systems diverges according to the orientations of the transmitter and receiver. Details are provided in the following sections for different models, namely LoS and NLoS channels.

## 2.2. Indoor VLC Channel Modeling

The following subsection outlines various models applied for both LoS and NLoS indoor VLC channels. For impulse response analysis, we consider a room wall as a surface acting as a receiver, with the received power as given in Equation (4).

The propagation model is represented as follows:

$$h(t; E, R, \lambda) = h^{(0)}(t; E, R, \lambda) + \sum_{k=1}^{\infty} h^{(k)}(t; E, R, \lambda) \tag{5}$$

where $h^{(0)}$ is a component that represents the LoS and $h^{(k)}$ is k-reflection for a source with wavelength $\lambda$ and a system with R receivers.

Using ray analysis, considering the second reflection, we get the received power as

$$P_\tau^{(2)} = P_t \, {}^\rho/_{10} \tag{6}$$

where received power is a product of reflection off the walls due to 10 rays (randomly chosen) and $\rho$ is the reflection coefficient of the surfaces or walls.

The probability density function of the rays is given as

$$\frac{m+1}{2\pi} \cos^m(\alpha) \tag{7}$$

where m is the Lambertian order and $\alpha$ is the angle between the z-axis and a random ray vector. As the rays propagate around the room and bounce off the walls with a probability density function in a particular direction, this is summed up to obtain the impulse response.

### 2.2.1. Impulse Response for Line of Sight (LoS) Model

Considering E emitters and R receivers placed at a distance $d_{0,R}$, the impulse response of LoS is given as

$$h^{(0)}(t; E, R, \lambda) = \frac{1}{(d_{0,R})^2} R(\varphi, n, \lambda) A_{eff}(\psi) \delta(t - \frac{d_{0,R}}{c})$$ (8)

with $R(\varphi, n, \lambda)$ being the Lambertian model of the emitter and $A_{eff}$ the surface area of the receiver for collecting the signal [39].

$$A_{eff}(\psi) = \cos(\psi) rect(\frac{\psi}{FOV})$$ (9)

### 2.2.2. Impulse Response for Non-LoS Model or Multiple Bounces

In an environment containing reflectors, the receiver captures signals after multiple reflections. The distribution of reflections over the simulation time is considered as the $\rho(\lambda)$ reflection coefficient, as received by the receiver. The power after each reflection is given, as discussed in [40]:

$$P_{i,k}(E, R, \lambda) = \frac{1}{(d_{k,R})^2} R_S(\theta_{k,R}, \theta', \lambda) A_{eff}(\psi_{k,R})$$ (10)

Unlike the Lambertian model, here we use Phong's model, which has specular and diffuse components.

$$(\theta_{k,R}, \theta', \lambda) = \rho_k(\lambda) P_{inc}(\lambda) [\frac{r_d(k)}{\pi} \cos\theta_{k,R} + (-r_d(k)) \frac{2m(\lambda) + 1}{2\pi} \cos^m(\theta_{k,R} - \theta')$$ (11)

In the given model, we have a reflection coefficient, $\rho_k(\lambda)$, along with incident power, $P_{inc}$, and the percentage of the signal that is reflected diffusely, $r_d(k)$.

Power due to $N$ different rays is given as the incident power, where the product of the reflective coefficient is given as

$$P_{inc}(\lambda) = \frac{P_E(\lambda)}{N} \prod_{j=1}^{k-1} \rho_k(\lambda)$$ (12)

The total power for $N$ rays of reflections using the Dirac delta with time instances

$$t_{i,k} = \sum_{j=1}^{k} \left( \frac{d_{j-1,j}}{c} \right) + \frac{d_{k,R}}{c}$$ (13)

gives us multiple bounces or reflections, which are given as follows:

$$\sum_{k=1}^{\infty} h^k(t; E, R, \lambda) = \sum_{i=1}^{N} \sum_{k=1}^{K} P_{i,k}(E, R, \lambda) . \delta(t - t_{i,k})$$ (14)

The summation on the left-hand side is for $N$ reflection and the multiplication of the Dirac delta and power after reflections from the surfaces.

Substituting Equations (8) and (13) in Equation (7), we get the total impulse response for the rays of reflection:

$$h(t; E, R, \lambda) = \frac{1}{\left(d_{0,R}\right)^2} R_E(\varphi, n, \lambda) A_{\text{eff}}(\psi) \delta(t) + \sum_{n=1}^{M-1} P_n(E, R, \lambda) \delta(t - n\Delta t) \tag{15}$$

## 3. Power Measurement for Both LoS and NLoS Models

Considering LEDs as emitters, modeled as Lambertian radiation for a collection of wavelengths with axial symmetry, a model number n for each lobe of radiation is expressed as follows.

$$R_E(\varphi, n, \lambda) = \frac{n+1}{2\pi} P_E(\lambda) \cos^n(\varphi) \tag{16}$$

In the above equation, the nominal power $P_E$ is obtained by integrating $P_E(\lambda)$ over the wavelength intervals. The value of $\varphi$ is $-\frac{\pi}{2} \leq \varphi \leq \frac{\pi}{2}$. However, regarding power estimation based on the rays, which propagate around the room and bounce off the walls with a probability density function in a particular direction, his is summed up to obtain the impulse response. The total impulse response is composed of the primary and secondary reflections, with the amplitude varying upon reflections received in X, Y, and Z planes. Figure 1a,b show the spread of energy or the received power in the Y direction and the surface map of the reflections.

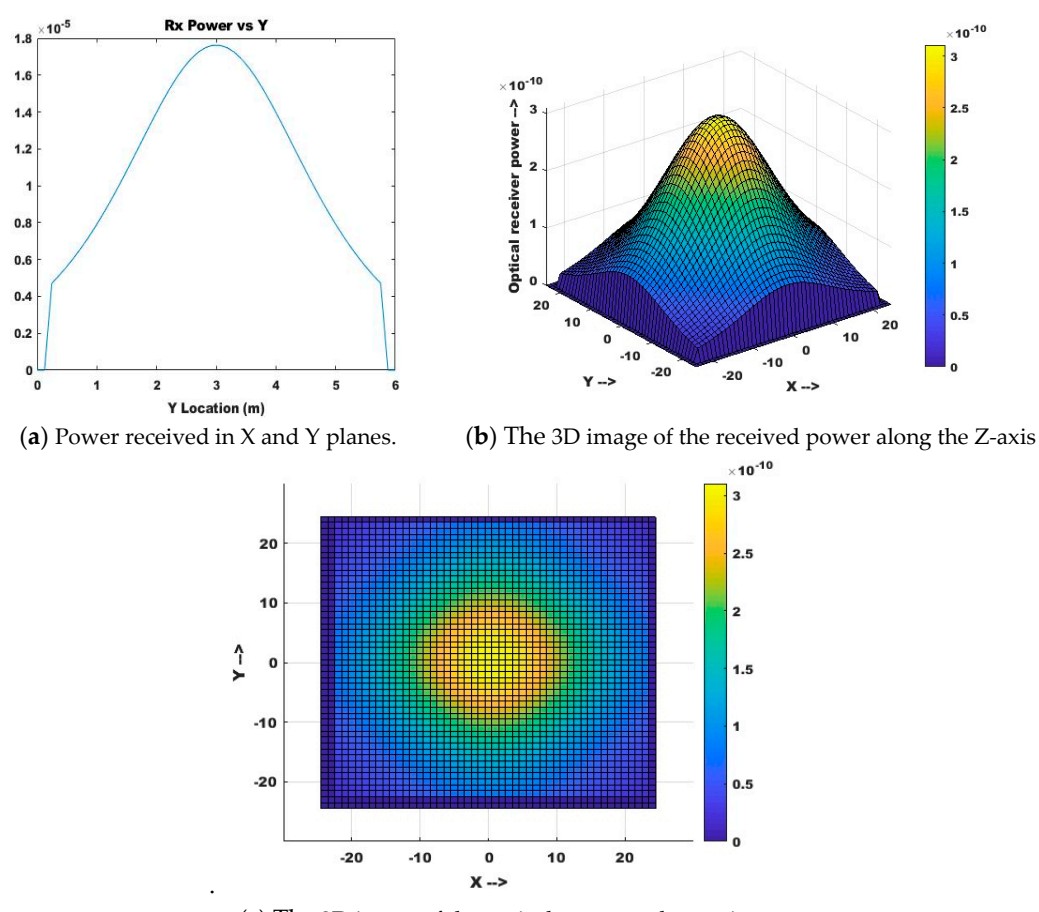

(**a**) Power received in X and Y planes.  (**b**) The 3D image of the received power along the Z-axis

(**c**) The 2D image of the optical power at the receiver.

**Figure 1.** Distribution of received power in each plane.

Figure 1a has no reflection state and the corresponding RSS is measured in the (X,Y) plane. The Z-direction in the RSSI measurement and (X,Y) are given for interpolation of points, as shown in Figure 1b. Figure 1c expresses the 2D presentation of the optical power at the receiver. Typically, the

positions of LEDs are parallel to the receiver plane and share a common clock, which transmits their three-dimensional coordinate information. The impetus for this research comes from the analysis of positional error using PDoP impulse response. It can be empirically shown that the signal-to-noise ratio (SNR) at the center of the room is more pronounced against the reflections. Hence, the effect of the reflections is small and weak at the center and does not contribute to the positioning accuracy.

Moreover, the light reflections of LEDs are taken into account (Single, 10 reflections). Figure 2 below shows the reflected or bounced signals measured after a single reflection and 10 reflections, respectively. As seen, the reflections cause changes in the received power, demonstrating a positional error.

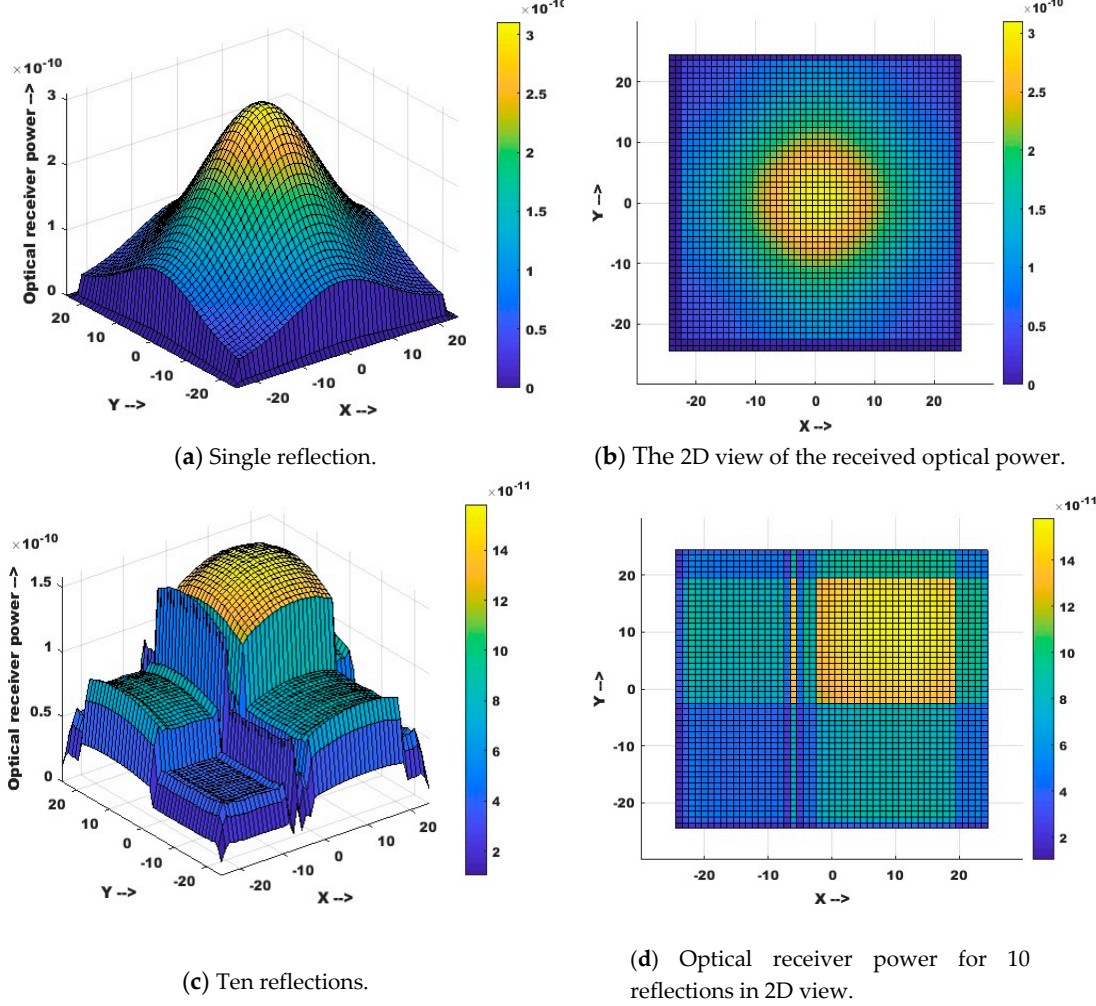

(**a**) Single reflection.

(**b**) The 2D view of the received optical power.

(**c**) Ten reflections.

(**d**) Optical receiver power for 10 reflections in 2D view.

**Figure 2.** Analysis of a single reflection and 10 reflections.

The 2D views of the single reflection and ten reflections for the power spread are shown in Figure 2b and Figure 2d. The power is focused in the first quadrant, as seen in Figure 2d while Figure 2b illustrates a more uniform spread. In order to visualize a uniform spread rather than a cross-sectional view we scaled the room dimensions from 6 × 6 to 24 × 24. This represents a scaled version of the room dimensions for visualization purposes.

From Figures 2, it can be seen that the power intensity distribution becomes more uniform at the edges and corners. As reflections increase the received power is reduced, creating an error in positioning. It can be seen that positioning error across the room is low, especially in the corner. The error is pronounced because of the low SNR. Comparing Figure 2a and Figure 2b, we find that the

reflections have a lower SNR, as seen in the concentration between (0,1) values. The receiver plane is closer to the floor, hence the improvement in determination accuracy. Comparing the single reflection versus ten reflections shows the spread in energy in the (X,Y) plane, as seen in Figure 2a,b. The spread of energy is non-uniform in the case of higher reflections, and hence is useful for accuracy calculation.

## 4. Adaptive Collaboration Positioning

As discussed previously, the DoP is used to illustrate the effect of geometry on the relationship between measurement error and position determination error. DoP has various components, such as PDoP and geometric DoP. PDoP is a subset of DoP, where the precursor RSS value for the DoP is analyzed for positional estimates. The proposed positioning algorithm for PDoP at the receiver or photodiode is shown in Figure 3.

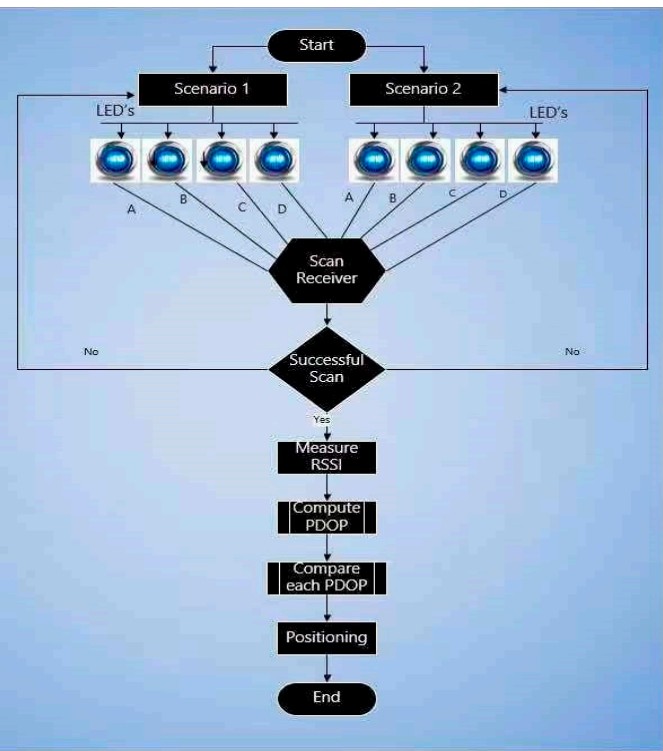

**Figure 3.** Schematic representation of the positioning algorithm based on positional dilution of precision (PDoP). Note: RSSI = received signal strength indicator.

These PDoP factors are defined as the ratio of the statistical value of position errors to the standard deviation (SD) of measurement errors. In this paper, we focus on PDoP, which is defined as:

$$PDoP = \frac{\sqrt{\sigma_x^2 + \sigma_y^2 + \sigma_z^2}}{\sigma_{UERE}} \tag{17}$$

Here, UERE (User-Estimated Range Error) and is used to infer PDoP for RSS-based localization using VLC. Considering m LEDs and one receiver in the localization premises, the measuring equation for the ith LED is

$$g_i = \frac{k_i}{r_i} + n_i \tag{18}$$

where *k* is equal to

$$\frac{m+1}{2} RnookPLEDA\, cos^m(\phi)Ts(\Psi)g(\Psi)\, cos(\Psi) \tag{19}$$

where $m$ is the Lambertain order, which is given by $\quad m = \frac{-ln}{cos\frac{\phi i}{2}}\quad$ here, $\varnothing$ is the radiation angle with

respect to each LED and n is zero-mean Gaussian distribution.

Here, $r$ is the distance between the ith LED and the receiver, which is given by

$$r_i = \sqrt{(x_i - x)^2 + (y_i - y)^2 + (z_i - z)^2} \tag{20}$$

where $(x,y,z)$ denotes the coordinates of the receiver in three-dimensional (3D) space, and $(x_i, y_i, z_i)$ denotes the coordinates of ith LED in 3D space.

A linear equation can be written in a matrix form as follows:

$$\Delta g = H\, \Delta X \tag{21}$$

where

$$\Delta g = \begin{pmatrix} g_1 - \hat{g}_1 \\ g_2 - \hat{g}_2 \\ \vdots \\ g_M - \hat{g}_M \end{pmatrix} \quad H = \begin{bmatrix} a_{x1} & \cdots & a_{z1} \\ \vdots & \ddots & \vdots \\ a_{xM} & \cdots & a_{zM} \end{bmatrix} \quad \Delta X = \begin{pmatrix} \Delta x \\ \Delta y \\ \Delta z \end{pmatrix} \tag{22}$$

When the least squares are used (n ≥ 3), we can obtain the estimation of $\Delta X$ from the following equation

$$\Delta X = (H^T H)^{-1} H^T\, \Delta g \tag{23}$$

where H is the RSSI measured from impulse response. H is not square, but $(H^T H)^{-1}$ is a non-singular and positive definite matrix.

When the measurement noise is taken into consideration, we have

$$\Delta X + dx\ =\ (H^T H)^{-1} H^T\, (\Delta g + n) \tag{24}$$

where $n$ is the measurement noise vector n = $(n_1, n_2, \ldots n_m)^T$, along with the position error vector, dx = $(dx, dy, dz)^T$. Considering the noise vector, here all 'i' are independent and identically distributed (i.i.d.) random variables, and each $n_i$ is subject to zero-mean Gaussian distribution. Therefore, each $dx_i$ is also subject to zero-mean Gaussian distribution.

We obtain the covariance of the positional vector and covariance of the noise vector:

$$cov(dx) = E[dxdx^T] = E[(H^T H)^{-1} H^T n\, n^T H(H^T H)^{-1}] = (H^T H)^{-1} E(nn^T) \tag{25}$$

$$E[n_i n_j] = \begin{cases} \sigma_n^2 \text{ if i = j} \\ 0 \text{ if i} \neq \text{j} \end{cases} \text{ i,j = 1,2} \ldots \text{m} \tag{26}$$

where $\sigma_n^2$ is the variance of the measured noise.

The covariance of position error vector dx can be rewritten as

$$cov(dx) = (H^T H)^{-1}\, I_{mxm}\, \sigma_n^2 = (H^T H)^{-1} \sigma_n^2 \tag{27}$$

Considering the left-hand side of the equation

$$cov(dx) = \begin{bmatrix} \sigma_x^2 & \sigma_{xy}^2 & \sigma_{xz}^2 \\ \sigma_{yx}^2 & \sigma_y^2 & \sigma_{yz}^2 \\ \sigma_{zx}^2 & \sigma_{zy}^2 & \sigma_z^2 \end{bmatrix} \tag{28}$$

Comparing the principal diagonal elements

$$\sqrt{\sigma_x^2 + \sigma_y^2 + \sigma_z^2} = \sqrt{T_r(H^T H)^{-1}}\, \sigma_n \tag{29}$$

where $\sigma_x^2, \sigma_y^2, \sigma_z^2$ are the variances of position error in x,y,z dimensions. Therefore, the expression of PDoP for the RSS-based positioning system is given by

$$PDOP = \sqrt{T_r(H^T H)^{-1}} \tag{30}$$

where $T_r$ denotes the trace operation of the positioning matrix.

The accuracy of the proposed scheme is based on the orientation of the receiver and densities of the fixture lights. The role of the reflective co-efficient is significant in measuring the intensity. The following section comprises the simulation environment and results.

## 5. Simulation Setup, Results, and Discussion

In this section, the experimental parameters and their results are considered. Here, the best configuration and effectiveness of the proposed system are evaluated under two scenarios. The RSS values for a VLC network along with multiple reflections are evaluated for each scenario. The proposed system is validated using Communication and Lighting Emulation Software (CandLES) [41]. CandLES uses the MATLAB software interface.

Furthermore, the RSS values that were generated for single and multiple reflections were simulated. The software tool has modulation, transmitters, optics, channel noise, interference, receivers, and decoding modules. Parameters such as the room size, the orientation of transmitters and receivers, wall reflectivity, and obstacles were taken into account. The performed experiments were focused on comparing positioning error. The system robustness and computation time were taken into consideration. The results obtained from simulations were adequately interpreted to analyze the channel model. The obstacle-free model was used as a transmission medium, and in order to calculate RSS values, the zero reflection was used, which signifies LoS and the existence of reflections (NLoS). The receiver was configured by using the value of 90° for field of view (FOV) and a photosensor area of 100 mm². Finally, the reflectivity (%) of the wall, ceiling, and floor were set to 0.8, 0.35, and 0.60, respectively. All experiments were carried out on an AMD FX™-8300 Eight-Core Processor with 3.30 GHz/8 GB RAM on a non-dedicated Windows machine. As mentioned, CandLES was used to build an RSS dataset to compute the impulse response and obtain the dimensions of a room. Both components of the optical transmission, namely direct LoS and NLoS, were taken into account in a 6 × 6 × 6 m room. A direct component with 10 reflections was taken into account with given parameters. The experiments were centered on comparing the accuracy and positioning error, and the system robustness and computation time were taken into consideration. The parameters specification for simulation against scenarios 1 and 2 are listed in Table 1 [42,43].

**Table 1.** Simulation parameters.

| Parameters | Values |
|---|---|
| Room Size | 6 m × 6 m × 6 m |
| Reflectance ($\varrho$) of ceiling | 0.35 |
| Reflectance ($\varrho$) of wall | 0.8 |
| Reflectance ($\varrho$) of floor | 0.60 |
| Coordinates of LED Sources for Scenario 1 | A (0,0,6), B (6,0,6) C (0,6,6), D (6,6,6,) |
| Coordinates of LED Sources for Scenario 2 | A (2,2,6), B (4,2,6) C (0,6,6), D (6,6,6) |
| Transmitted power | 16 w |
| Center luminous flux | 300 lm |
| Height of LED source | 5.15 m |
| Receiver detection area | 100 m² |
| Height of receiver | 0.085 m |
| FOV of receiver | 70° |
| Elevation of receiver | 90° |

Note: LED = light-emitting diode; FOV = field of view

LED transmitters were installed on the ceiling to transmit data. The LEDs transmit data uniformly without any central unit controlling them and no collaboration allowed between them. The relative positions of LEDs and receivers with respect to scenarios 1 and 2 are shown in Figure 4a,b, respectively.

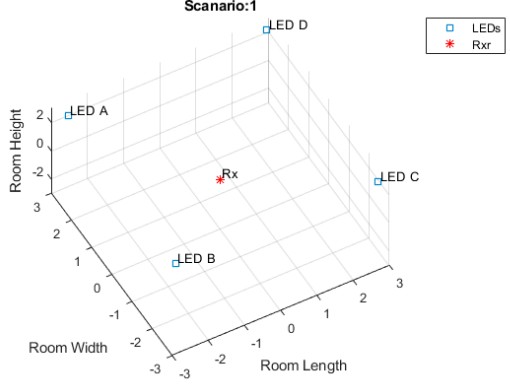

(**a**) Scenario 1 LEDs and receiver orientation.

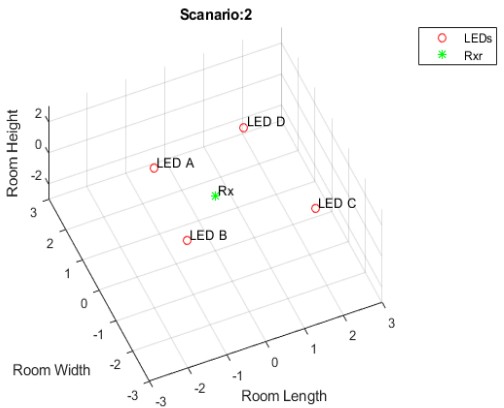

(**b**) Scenario 2 LEDs and receiver orientation.

**Figure 4.** Different orientations of transmitters and receivers for different scenarios.

Two indoor localization scenarios using VLC are presented. In scenario 1, the room dimension L × W × H is 6 × 6 × 6 m, and four LED lamps are located at the four corners of the ceiling, respectively, as shown in Figure 4a. The position of four LED lamps in scenario 1 is denoted by A (0,0,6), B (6,0,6), C (0,6,6), and D (6,6,6). In scenario 2, two LEDs are located in the room corners and two others are located close to the center of the room (coordinates), as shown in Figure 4b. The positions of four LED lamps in scenario 2 are denoted by A (2,2,6), B (4,2,6), C (0,6,6), and D (6,6,6) (verified experimentally).

Figure 5a,c demonstrates the spread and focusing of power over the volume of the room and the proximity of the sources, while Figure 5b,d presents the 2D view of received optical power for scenarios 1 and 2, respectively. In each scenario, we chose 31 × 31 positioning nodes on the Z = 1 plane, with each node fixed at intervals; aside from its neighboring nodes, the receiver axis is parallel with Z-axis. Figure 5a,b illustrates the energy spread between the LED grids of two different configurations. There is a reduction of energy in the LED grids, which is laid out in a far grid against the closer one. We calculated the PDoP value for each positioning node. The statistics of all the PDoP values calculated for 31 × 31 nodes are analyzed. Using the least squares estimation for the position, as shown in Equation (20), we calculated the error matrix. The resultant matrix provides a solution that reduces the sum of squares of the residuals or errors. We collected the residuals in a vector that was used to compute the cumulative distributive function of the Student's t-distribution. The

residuals or error were used to calculate the RMSE for positional changes. The RMS delay spread parameter is the only metric that relates to positional accuracy based on the impulse response. The spatial coordinates are Cartesian products, while the LED luminous flux is in the spherical coordinates; since it is considered to be a solid angle, the delay spread variation gives the positional variations. Appendix A shows the analytical method of RMS delay spread and measured luminous flux. Here, we have the power distribution in the form of t-statistics, with "t" giving the relative error in comparison to the null hypothesis. The "p-value" provides the statistical significance of measured values against theoretical values.

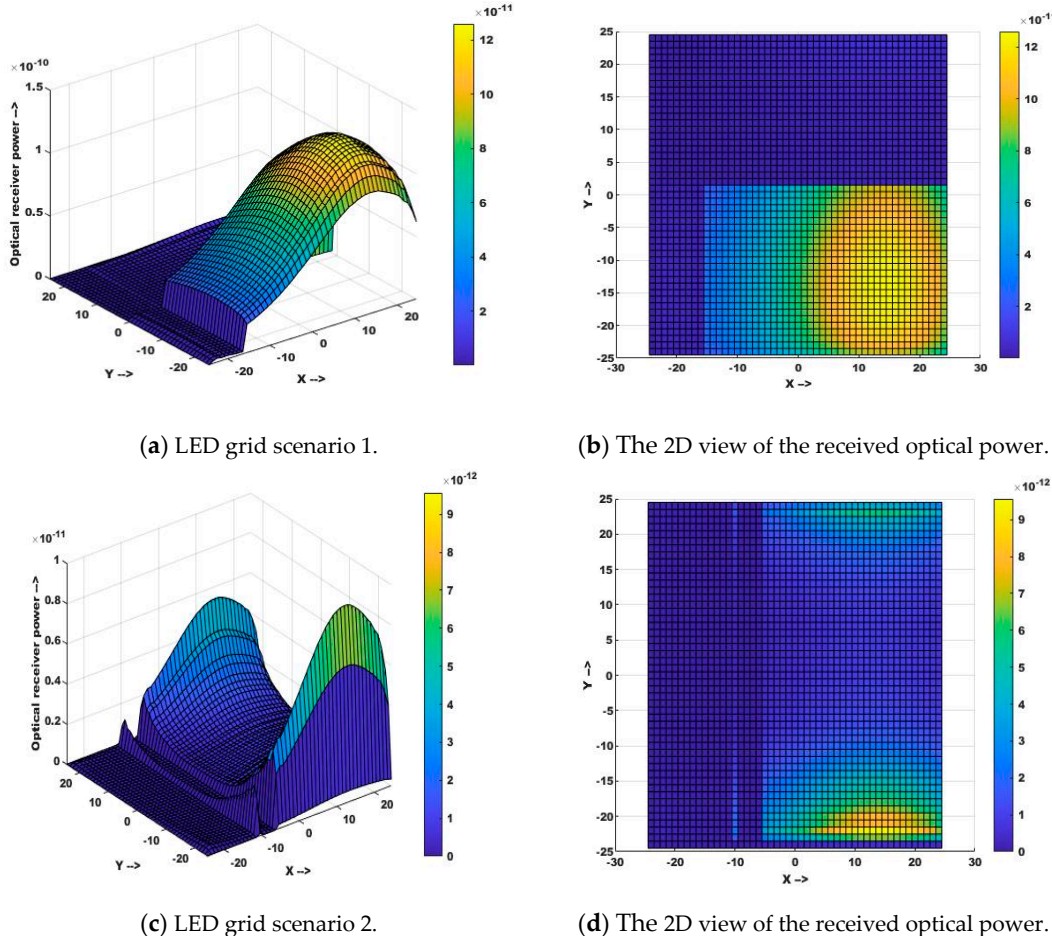

(**a**) LED grid scenario 1.

(**b**) The 2D view of the received optical power.

(**c**) LED grid scenario 2.

(**d**) The 2D view of the received optical power.

**Figure 5.** Proximity of light sources.

As seen in Table 1, for our 2 different scenarios, we calculated the sum of square errors, *t*, *p*, and RMSE. We considered the power from LED-A as the reference power value, with corresponding *t* and *p*-values. Tables 2 and 3 provide the SSE, t, p, and RMSE values for both cases. The RMSE value of $1.4054 \times 10^{-6}$ and $1.13 \times 10^{-6}$ is a comparison considering the overall volume of room dimensions, which is enhanced by 24%, as mentioned in Appendix A.

**Table 2.** Scenario 1. Least squares estimation of PDoP values for received signal strength (RSS) localization. RMSE = root means square errors.

| Parameters | LED-A | LED-B | LED-C | LED-D |
|---|---|---|---|---|
| SSE | 0 | 0.3492 | 0.1845 | 0.2814 |
| *t* | −2.4217 | −1.6456 | 0.8489 | 2.5946 |
| *p* | 0.0193 | 0.1064 | 0.4002 | 0.0125 |

| RMSE | $1.4054 \times 10^{-6}$ |
|------|--------|

**Table 3.** Scenario 2. Least squares estimation of PDoP value for RSS localization.

| Parameters | LED-A | LED-B | LED-C | LED-D |
|------------|-------|-------|-------|-------|
| SSE | 0 | 126.4352 | 0.0646 | 168.92 |
| *t* | −1.3585 | −2.3785 | 17.8397 | 2.3303 |
| *p* | 0.1807 | 0.0214 | 0 | 0.024 |
| RMSE | $1.13 \times 10^{-6}$ | | | |

Finally, proportional references are cited for comparison purposes in terms of cost, availability, energy efficiency, and complexity. As can be seen in Table 4, the performance of this work is far better than the other state of the art indoor localization techniques involving DoP metrics.

**Table 4.** Comparative evaluation in terms of application of dilution of precision (DoP) metrics in indoor positioning.

| References | DoP (Indoor Positioning) | Evaluation of Frame Work | | | |
|------------|--------------------------|------|--------------|-------------------|------------|
| | | Cost | Availability | Energy Efficiency | Complexity |
| [44] | GDoP | High | Low | Low | High |
| [45] | DoP | Moderate | High | Low | High |
| [46] | HDoP, VDoP | High | Low | Moderate | Moderate |
| This work | PDoP | Low | Moderate | High | Low |

Note: VDoP = vertical DOP; GDoP = geometric DOP; PDoP = positional DOP; HDoP = horizontal DOP

## 6. Conclusions

This paper presents a clear and systematic investigation of the penetrating VLC channel models for LoS and NLoS in terms of an impulse response. This study is performed for the indoor VLC channel, which is beneficial with a highly performing sub-system comprising a well-analyzed and novel localization algorithm. The positioning errors around the room are compared using two scenarios. The results have shown that the edge or corner regions reduced the impact of reflections, demonstrating the focus of power across the room. We have proven that PDoP is the decisive factor that helps the system in reducing eclectic localization errors for RSS-based indoor localization. This is done by employing the VLC with modified LED patterns and quantifies the consequence of LED configurations on the position error. It has been observed that the elements that form an appropriate geometry are capable of providing optimized accuracy. Further, the model output demonstrates both SSE and RMSE among the positioning nodes in both scenarios. The values in scenario 1 are greater than those in scenario 2, and the residuals of LEDs are kept as reference values. Surprisingly, the measured power resulting in RMSE shows that the dependency of neighboring power values is greater. Hence, it can be concluded that the orientation of LEDs quantizes the spread of power, which affects accuracy errors. The simulation results demonstrate that the observations and conclusions of this work can influence the design and implementation of future optical wireless systems. Moreover, future work will investigate of the angle of radiation and angle of incidence, while taking the other dilution factors, including VDoP and HDoP, into account for complex indoor environments.

**Author Contributions:** Writing—original draft preparation, methodology, software, and formal analysis, M.I.; conceptualization, validation, data curation, and visualization, M.I., J.A., A.M., and M.N.S.; supervision, resources, project administration, and funding acquisition, W.L.; investigation, W.L., M.I., J.A., A.M., and M.K.; writing—review and editing, J.A., A.M., M.K., and M.M.U.

**Funding:** This work was supported in part by the National Nature Science Foundation of China, grants number: 61672448.

**Acknowledgments:** The authors would like to give thanks to Michael Brandom Rahaim for the CandLES software.

**Conflicts of Interest:** The authors declare no conflict of interest.

## Appendix A

RMS delay spread is a measure and channel spread is caused by multipath propagation, including inter-symbol interference (ISI). It is a predictor of SNR and is used to remove the dependency on channel impulse response. Here h(t) is given as the RMS delay [44]:

$$\tau_{RMS} = \sqrt{\frac{\int_{-\infty}^{\infty} (t-\tau_0)^2\, h^2(t)dt}{\int_{-\infty}^{\infty} h^2(t)dt}} \quad \tau_0 = \frac{\int_{-\infty}^{\infty} t\, h^2(t)dt}{\int_{-\infty}^{\infty} h^2(t)dt}$$

where t is the propagation time and $\tau_0$ is average delay time; h(t) and RMS are fixed when the transmitters, receivers, and reflectors are given.

Percentage of improvement = $((1.4054 - 1.13)/1.13) \times 100 = 24\%$.

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
