# Peer review of "A Novel Localization Technique Using Luminous Flux"

_applsci, doi:10.3390/app9235027_

Round 1

Reviewer 1 Report

Brief summary

Authors analysed positioning characteristics of indoor positioning system using VLC technology. Power of received signal is used to obtain distances to the LED light sources and eventually to determine receiver position. An impact of light reflections is considered during analysis. Numerical experiment was carried out to study light spreading out.

Broad comments

As VLC is a new positioning technique, the research subject touches up-to-date problems. Numerical modelling of studied phenomenon is also reasonable.

Description of the theoretical part has been considerably improved in comparison to the previous version of the manuscript, but it is still not clear in some places.

Specific comments

Line 42 and 46: 'i)' and 'ii)' should start from new line

Line 99 (and several other): GPS is only one of positioning systems. GNSS is better for scientific text

Figure 1: Additional explanation is necessary. What is the position of LED in relation to plane of analysis (Fig. 1c). Pictures b) and c) are to small.

Figure 2: All the pictures are too small. Title of picture c) font size needs to be adjusted.

Line 223-224: Sentence "For visualizing a uniform spread rather than a cross-sectional view, we have scaled the room dimension that is 6 x6 to 24x 24" is no clear. Please explain it.

Figure 3: "PDOP" should be changed into "PDoP"

Line 246: Formula should be placed in individual line (not in the text). Is k the same as ki in formula (17)? There is no i index in formula.

Line 255: We can't describe H matrix as non-singular (nor singular) because H is not sqare.

Adjustment is necessary in formulas (23) and (24).

Signs begining lines 265 and 267 are confusing. Please rearrange this fragment of text.

Table 1: Is there any reason in showing azimuth of the LED source/receiver when its elevation is ±90º ?

Figure 4: The LED-s positions seem not to correspond to their coordinates.

Line 312: In scenario 2 two LED-s are located in the room corners and two others close to the centre of the room (coordinates). We can't say "four LED lamps are located in the central area of the ceiling"

Figure 5: Pictures a) .. d) are indecipherable (too small)

Line 366: How do authors understand ".. orientation of LEDs is proportional to the spread of power .." it's totally unclear for me.

Conclusion:

The result of analysis is interesting and can be published but authors should revise the article and eliminate all the mistakes.

Author Response

Response to 1st Honorable Reviewer

Peer Reviewer Assessment

Brief summary: -

Authors analysed positioning characteristics of indoor positioning system using VLC technology. Power of received signal is used to obtain distances to the LED light sources and eventually to determine receiver position. An impact of light reflections is considered during analysis. Numerical experiment was carried out to study light spreading out.

Broad comments: -

As VLC is a new positioning technique, the research subject touches up-to-date problems. Numerical modelling of studied phenomenon is also reasonable. Description of the theoretical part has been considerably improved in comparison to the previous version of the manuscript, but it is still not clear in some places.

Authors’ Response

The authors are wholeheartedly thankful to the honourable reviewers for such a constructive comment for improving our manuscript. The authors have tried their best to cover all aspects of research content. The modification is also highlighted for the convenience. Furthermore, the authors have modified the manuscript according to the best of their knowledge. However, the further improvement if suggested by the reviewers will be highly encouraged and appreciated. Hopefully, the revised manuscript will be acceptable now.

Specific comments: -

Line 42 and 46: 'i)' and 'ii)' should start from new line.

Authors’ Response

The authors are wholeheartedly thankful to the honorable reviewer for technical improvements. The modification is done according to the direction on the page-2 line 41 and 45.

Line 99 (and several other): GPS is only one of positioning systems. GNSS is better for scientific text.

Authors’ Response

Thank you very much, honorable reviewer for such a constructive feedback and to point out such a critical point. The GPS is replaced with GNNS in the whole manuscript.

Figure 1: Additional explanation is necessary. What is the position of LED in relation to plane of analysis (Fig. 1c). Pictures b) and c) are to small.

Authors’ Response

Dear Sir, thank you very much for such a critical comment and for pointing out this imperative improvement in the paper. Typically, position of LEDs is parallel to the receiver plane sharing a common clock, transmit their three-dimension coordinate information. The information is included on the page-8, line. 206-207.Figure 1. (b & c) are revised with more clear formatting on page.7, Line. 199-203

Figure 2: All the pictures are too small. Title of picture c) font size needs to be adjusted.

Authors’ Response

The authors are wholeheartedly thankful to the honorable reviewer for such critical points for improving this manuscript all graphical representations are reconsidered according to the standard format and their font sizes are set according to the journal guideline. Page-8, Line.216-221.

Line 223-224: Sentence "For visualizing a uniform spread rather than a cross-sectional view, we have scaled the room dimension that is 6 x6 to 24x 24" is no clear. Please explain it.

Authors’ Response

We are very thankful to the reviewer for very critically going through the manuscript and highlighting the points. The cross-sectional view tends to a scaled version of the room dimensions for visualization purposes and its explanation in included in the manuscript on page 08 and 09. Line .224-226. 

Figure 3: "PDOP" should be changed into "PDoP"

Authors’ Response

Dear Sir, thank you very much for showing your deep reviews on the whole manuscript. The correction is made accordingly according to the suggestion on page-09, line.239 

Line 246: Formula should be placed in individual line (not in the text). Is k the same as ki in formula (17)? There is no i index in formula.

Authors’ Response

The authors are wholeheartedly thankful to the honorable reviewer for such critical points for improving this manuscript. The K depends on the radiation which varies with respect to radiation angle, formula is revised and placed in a sperate line with corresponding equation number. page-10, line.248-251. 

Line 255: We can't describe H matrix as non-singular (nor singular) because H is not square.

Authors’ Response

Dear Sir, thank you very much for such a critical comment and for pointing out this imperative improvement in the paper. H is the RSSI measured from impulse response. H is not square but,  -1 is a non-singular and positive definite matrix on page-10, line 257-259 and here it is (verified numerically). 

H is a rectangular matrix(50x3) but (HTH)-1 is a 3x3

Case 1:

31809989114.6071     -25576709365.4410    -3624213605.10001

-25576709365.4410    26115990681.4644     -344892243.756263

-3624213605.10001    -344892243.756263    2405677939.58107

Eigen values are:  1.0e+10 * 0.0047

                                                 0.5470

                                                5.4814

Case 2:

31811820952.3254     -31106666131.4527    -5515841940.18060

-31106666131.4527    31524418898.8966     1696956900.29032

-5515841940.18060    1696956900.29032     23959909796.0024

Eigen values are:

  1.0e+10 *    0.0251

                        2.3608

                        6.3438

Adjustment is necessary in formulas (23) and (24).

Authors’ Response

Thank you very much, honorable reviewer for such a constructive feedback and to point out such a critical point. Since the eigen values are positive and the (HTH)-1 is a square matrix our equations for calculating the PDoP is justified.

Signs beginning lines 265 and 267 are confusing. Please rearrange this fragment of text.

Authors’ Response 

Authors appreciate the positive improvements from reviewer. The signs have been removed according to the direction given by honorable reviewer on the page-11, line 265-270.

Table 1: Is there any reason in showing azimuth of the LED source/receiver when its elevation is ±90º ? 

 Authors’ Response

Thank you very much, honorable reviewer for such a constructive feedback and to point out such a critical element. The simulator CandLES has inbuilt setup for azimuth and elevation methods. However, to save the space the extra information is excluded from the table.1 page. 12.

Figure 4: The LED-s positions seem not to correspond to their coordinates.

Authors’ Response

The authors have made appropriate LED position correspond to their location in the figure.4 according to the guidance provided by the respectable reviewer on the page -13, line 306-307.

Line 312: In scenario 2 two LED-s are located in the room corners and two others close to the centre of the room (coordinates). We can't say "four LED lamps are located in the central area of the ceiling".

Authors’ Response

Thank you very much, honorable reviewer for such a constructive feedback and to point out such a critical point the changes are made according to the instruction given by honorable reviewer on the page-13, line- 313-314.

Figure 5: Pictures a) .. d) are indecipherable (too small)

Authors’ Response

Dear Sir, thank you very much for indicating the formatting improvement. The pictures are revised according to the journal standard format for your kind consideration on page 13 and 14 respectively. 

Line 366: How do authors understand ".. orientation of LEDs is proportional to the spread of power .." it's totally unclear for me.

Authors’ Response

Dear Sir, thank you very much for such a critical comment and for pointing out this imperative improvement in the paper. The orientation of LEDs is actually considered to quantize the spread power to evaluate accuracy error. The appropriate correction has been made in the manuscript on page-16, line number.367-368.

Reviewer 2 Report

The quality of the presentation related to Figures 1, 2, and 5 is still needed to be improved. Some subfigures are too small to figure their details out.

Author Response

Response to 2nd Honorable Reviewer

Peer Reviewer Assessment

Comments and Suggestions for Authors

The quality of the presentation related to Figures 1, 2, and 5 is still needed to be improved. Some subfigures are too small to figure their details out.

Authors’ Response

The authors are wholeheartedly thankful to the honourable reviewers for such a constructive comment for improving our manuscript. The authors have revised all the figures to the journal standard format for your kind consideration. However, further improvement if suggested by the reviewers will be highly encouraged and appreciated. Hopefully, the revised manuscript will be acceptable now.

This manuscript is a resubmission of an earlier submission. The following is a list of the peer review reports and author responses from that submission.

Round 1

Reviewer 1 Report

In the paper, the author claimed that the systematic investigation of a VLC channel was performed for both direct and indirect LoS and the position accuracy was enhanced by 24% by utilizing the PDoP metric.

The presentation related to positioning was not clear. The paper should clearly present how to estimate the coordinate (x, y, z) of the receiver in (18). In the abstract, the authors claimed that the position accuracy could be enhanced by 24% by utilizing the PDoP metric. But, it seems that there are no details available in sections 5 and 6 to support this claim. The results presented in the simulation section is not solid enough. It is recommended to design experimental work to validate the proposed positioning method. The authors did not compare the performance in general to other traditional positioning schemes, like the DoP at the same conditions of the experimental setup. Moreover, the quality of presentation is needed to be greatly improved. There are many typos, poor quality figures, and grammar error in the manuscript.

Reviewer 2 Report

Review Report

A Novel Localization Technique by using Luminous Flux

Brief summary

Authors analysed positioning characteristics of indoor positioning system using VLC technology. Power of received signal is used to obtain distances to the LED light sources and eventually to determine receiver position. An impact of light reflections is considered during analysis.

Broad comments

VLC is a new positioning technique and the research subject touches up-to-date problems. The idea of creating numerical modelling of studied phenomenon is also reasonable. Also the general idea of the numerical experiment is good.

Unfortunately the description of the theoretical part (sections 2, 3 and 4) as well as description of numerical simulations (section 5) is written very poorly.

Specific comments

Authors use large number of abbreviations. Some of them are commonly known (eg. GPS, LED) but some are specific for narrower areas but they are not explained (VLC, SITE). Some others are explained more than once (UWB) or explained badly (AOA stands for 'angle of arrival').

Some abbreviations are written in various ways (PDoP and PDOP, N-LoS, NLoS and Non-LOS, PDoP and PDOP)

Formulas:

- Conventionally we don't use quotation marks while explaining formula symbols.

- Some symbols are not explained at all (eg. h(t) in formula 1)

- Line 238: Where is K in formula (17)?

- What is the matrix H used i formulas (19) and (20)?

- Symbol Δx in formula (20) is used in two different meanings.

- Line 250: What n finally is? Please decide.

Figure 3: very poor quality of picture.

Figure 2: title does not explain what is presented in the pictures

Simulation:

Line 290: Table 1 does not come from publication [46].

Table 1:

- Room size is declared as 6x6x3 and LED's coordinates suggest 6x6x6

- What does "Height of reviever" mean?

Line 302: Where is Fig. 6 ?

Figure 5: The axes description are not described and the values do not coincide with room size.

Figures 1b, 2, 5 information very unclear. Why did authors decide to present 2D data in 3D chart?

Minor mistakes:

Formulas (26) and (27) - the square root symbol should cover also (HTH)-1.

Formula (9) - please adjust font size.

Conclusion:

The idea of analysis is worth presenting but authors should rebuild the article to eliminate all the mistakes and make it clearer.